# Influence of Texture on the Mechanical Properties of a Mg-6Al-1Zn-0.9Sn Alloy Processed by ECAP

**DOI:** 10.3390/ma14102664

**Published:** 2021-05-19

**Authors:** Hong Xu, Zhi-Peng Guo, Ping-Yu Zhang, You Zhou, Pin-Kui Ma

**Affiliations:** Key Laboratory of Automobile Materials of Ministry of Education, School of Materials Science and Engineering, Nanling Campus, Jilin University, No. 5988 Renmin Street, Changchun 130025, China; xh@jlu.edu.cn (H.X.); guozp18@mails.jlu.edu.cn (Z.-P.G.); zpy17@mails.jlu.edu.cn (P.-Y.Z.); zhouyou17@mails.jlu.edu.cn (Y.Z.)

**Keywords:** ECAP, microstructure evolution, weaken texture, mechanical properties, magnesium alloys

## Abstract

The microstructure and mechanical properties of a Mg-6Al-1Zn-0.9Sn alloy processed by equal channel angular pressing (ECAP) at temperatures of 250 °C and 300 °C were investigated. It was found that the refinement of the microstructure was very dependent on the processing temperature. The main reason for the difference in grain refinement was the precipitation of secondary-phase particles. Texture information obtained by electron back-scatter diffraction (EBSD) showed the gradual formation of a 45° texture during the ECAP process, while the maximum intensity was different for processing temperatures at 250 °C and 300 °C. By calculating the contribution from different strengthening mechanisms, it was found that a 45° texture had a huge influence on grain boundary strengthening and thus the yield strength.

## 1. Introduction

The advantages of their high specific strength give magnesium (Mg) alloys wide prospects for lightweight applications in the aerospace, automotive, and electronic fields [1,2]. However, the low formability and strong plastic anisotropy of Mg alloys limit the development of their wider industrial applications [3,4]. Now, some new types of Mg alloys have been developed based on the common Mg-Al-Zn system, which is by adding extra rare earth elements [5,6]. The rare earth elements (such as Nd and Ce [5,6]) can effectively ameliorate the formability of Mg alloys at room temperature (RT). Compared with expensive rare earth elements, adding the inexpensive Sn element to Mg alloys can improve not only their creep resistance, but also the mechanical strength and toughness at RT [7,8].

As is well known, microstructure refinement is also one of effective ways of improving the mechanical properties of alloys [9]. Thus, many research works have been devoted to achieving microstructure refinement by thermo-mechanical processes, such as extrusion and rolling [10,11]. Nevertheless, the microstructure refinement obtained by this processing is not enough to greatly improve the mechanical properties of magnesium alloys. Therefore, obtaining better microstructure refinement by proper processing is still the research focus of Mg alloys [12,13,14].

Severe plastic deformation (SPD) techniques have been proven to be effective in producing an ultra-fine microstructure, such as high-pressure torsion (HPT), asymmetric rolling (ASR), and ECAP [15,16,17,18,19,20]. In recent years, some research has been carried out on processing Mg alloys by ECAP [15,17,19]. Overall, the researches on Mg alloy ECAP include two aspects: microstructure refinement and texture modification [21,22,23,24]. Although ECAP can refine grains, the high strength and toughness may not be necessarily achieved. The yield strength of Mg alloys may be reduced after ECAP, especially those with strong basal textures [17,25,26,27,28,29]. How to ensure the reasonable match between microstructure refinement and texture modification during ECAP is the critical point to improve strength and toughness. Therefore, exploring the influence of microstructure refinement and texture modification on the mechanical properties of ECAP-processed Mg alloys is essential.

Although many research works on ECAP-processed Mg alloys have been reported, few studies have systematically analyzed the relationship among their microstructure, texture, and mechanical properties. Thus, the present work studied how the deformation temperatures (250 °C and 300 °C) affect the grain refinement, precipitation, texture evolution, and mechanical properties of an ECAP-processed Mg-6Al-1Zn-0.9Sn alloy.

## 2. Experimental Procedure

The 10 mm thickness extruded Mg-6Al-1Zn-0.9Sn (wt. %, ATZ611) plate was used in this work. Before EACP, the extruded ATZ611 plate was solutionized at 420 °C for 20 h, then quenched. The as-solutionized ATZ611 plate was cut into 10 mm × 10 mm × 80 mm bar-shaped ECAP samples along the extrusion direction. The ECAP samples were deformed through a 90° ECAP die using route Bc. Route Bc is when the next pass is rotated 90° in the same direction relative to the previous pass. When ECAP processed by this die, the single-pass strain ε =1.1. Considering that when the deformation temperature is lower than 250 °C, the samples will be broken in the ECAP process and the deformation temperature is too high to achieve the purpose of grain refinement, the deformation temperature of ECAP in this paper was selected as follows: 250 °C and 350 °C. Before ECAP, the temperature of the ECAP die was heated to the desired temperatures (250 °C and 350 °C) and maintained for 30 min. Then, the ECAP sample was placed in the ECAP die for 15 min to reach the deformation temperatures (250 °C and 350 °C) as well. A self-made temperature control system was used to control the temperature. For convenience, the deformed samples after X passes are referred to as the XP samples. The schematic of the ECAP process is shown in Figure 1a, where the ECAP sample was extruded from the vertical channel into the horizontal channel under the action of the punch rod.

Microstructure characterization was conducted by scanning electron microscopy (SEM: ZEISS EVO18, Oberkochen, Germany) and electron backscatter diffraction (EBSD: Oxford NordlysNano, Oxford, UK.). Phase analysis was performed by an X-ray diffractometer (XRD: D/Max 2500PC Rigaku, Tokyo, Japan), using a Cu target Kα-ray (λ = 1.5406Å). RT tensile tests were implemented using a SHIMADZU AGS–100kN machine (Tokyo, Japan), with a stain rate of 1 × 10^−3^ s^−1^. The 10 mm × 4 mm × 1.5 mm tensile samples were cut along the ED (Figure 1b).

## 3. Results and Discussion

### 3.1. Microstructure Evolution During ECAP

IPF maps and a {0001} pole figure of the as-solutionized ATZ611 alloy are shown in Figure 2. The as-solutionized ATZ611 alloy exhibited a uniform equiaxed grain structure delineated by well-defined boundaries, with d_ave_ of ~26.2 μm. The {0001} pole figure indicated a strong basal texture (Figure 2b) with a maximum intensity of 20.9 mr, which was analogous to the typical basal textures of extrusion plates reported in previous studies [30,31].

Figure 3 shows IPF maps of 250 °C–ECAP samples and 300 °C–ECAP samples. Their corresponding d_ave_ are shown in Table 1. After one pass (Figure 3a), a heterogeneous grain structure was formed in the 250 °C–1P sample, with the d_ave_ decreased to 6.2 μm compared to the as-solutionized ATZ611 alloy. Nearly 70% of the grains had sizes less than 5 μm. The formation of heterogeneous grain structures at 1PECAP of the ATZ611 alloys agreed with the results of Figueiredo et al. [17]. If the d_ave_ before ECAP were higher than a critical size, the refinement would start inhomogeneously with fine grains forming at the original grain boundaries for a few ECAP passes. For the ECAP-processed AZ31 alloy, the critical size for the formation of a bimodal size distribution was between 3 and 9 μm. Therefore, the formation of bimodal microstructures after one pass of ECAP in Figure 3a was because of the initial large grain size (~26 μm). As shown in Figure 3b, the size distribution of 250 °C-2P samples became more homogeneous, and d_ave_ was measured to be ∼3.6 μm (Figure 3e). After four passes of ECAP, grains became further refined, with d_ave_ being ~1.9 μm.

Compared with the 250 °C sample, it was clear that there were no elongated coarse grains in the 300 °C–1P sample (Figure 3d). The d_ave_ is ~16.5 μm, which was larger than that of the 250 °C–1P sample. With further increasing the ECAP passes, the grain structure became more uniform. However, it was quite clear that higher deformation temperature led to slow grain refinement during ECAP, resulting in larger grains.

XRD photos of the as-solutionized sample and ECAPed samples deformed at 250 °C and 300 °C are shown in Figure 4. It indicates that after the solution’s heat treatment, nearly all the secondary phase particles in the ATZ611 alloy dissolved into the α-Mg matrix, while a few Al_8_Mn_5_ phases remained. After one pass of ECAP at 250 °C, the diffraction peaks of the Mg_17_Al_12_ phase appeared, with the highest peak intensity at 2θ = 36.74°. During ECAP, dynamic precipitation of the Mg_17_Al_12_ phase was due to the increasing dislocations, and vacancies could accelerate the diffusion of alloying elements [32,33]. For the 250 °C–2P samples, the intensity of the Mg_17_Al_12_ diffraction peaks increased, and the diffraction peaks of the Mg_2_Sn phase appeared.

In contrast, for the 300 °C–1P sample, the peak intensities of the Mg_17_Al_12_ phase were lower, which may be due to the faster recover of dislocations and vacancies. Moreover, obvious diffraction peaks of the Mg_2_Sn phase could not be seen for the 300 °C–2P sample. It was proven that the precipitation of the Mg_2_Sn phase could nucleate at the edge-tip of the Mg_17_Al_12_ phase [34]. If the precipitated amount of the Mg_17_Al_12_ phase were small, then the precipitation of Mg_2_Sn would be small as well.

Figure 5 shows SEM images of samples deformed at 250 °C, with the d_p_ and f_p_ of the secondary-phase partials summarized in Table 1. Combined with the XRD in Figure 4, Al_8_Mn_5_ particles with sizes of 5–10 μm could be observed in the 250 °C–1P sample (Figure 5a). Many ~400 nm spot-shaped Mg_17_Al_12_ phases precipitated in the area of fine grains, while almost no phase particles were inside the elongated deformed grains. Upon increasing the ECAP passes to two, the particle distribution became more uniform, and many Mg_17_Al_12_ particles were located at the grain boundaries, while large Al_8_Mn_5_ particles were broken (Figure 5b). After four passes, no large Al_8_Mn_5_ particles remained (Figure 5c). During ECAP deformation, fine fragmented Al_8_Mn_5_ and sub-micron Mg_17_Al_12_ particles could pin the boundaries and hinder the growth of recrystallized grains effectively [35,36].

Similarly, Figure 6 shows the SEM images of samples deformed at 300 °C. For the 300 °C–1P sample, secondary-phase particles with f_p_ of 5% precipitated. With the ECAP passes increasing, the f_p_ of the secondary-phase particles decreased gradually. As shown in Figure 6c, a few Mg_17_Al_12_ phases with a d_p_ of 350 nm precipitated at the grain boundaries in the 300 °C–4P sample. Compared with the 250 °C samples, less particles precipitated (Figure 6a–c), which led to a larger grain size for the samples deformed at 300 °C.

Fractions of recrystallized grains in ECAP-processed samples were evaluated from the EBSD results (Figure 7), in which grains with misorientation angles less than 2°, between 2° and 15°, and above 15° were defined as deformed grains, substructured grains, and recrystallized grains, respectively. In the 250 °C–1P sample, most grains were substructured grains (Figure 7a). Upon increasing the ECAP passes to two and four, more fully recrystallized grains formed. For samples deformed by ECAP at 300 °C, the fractions of recrystallized grains were much higher compared to the 250 °C samples. Normally, the values of critical resolved shear stress (CRSS) for the activation of non-basal slip at RT are much larger than those for basal slip. It was found that upon increasing the temperature over 200 °C, the non-basal dislocation was energetically more favorable to move. In addition, adding the Sn element to Mg alloys can also promote the activity of non-basal slip [7]. Furthermore, compared to basal dislocations, the non-basal dislocations have a higher probability to lock with each other and then form dislocation tangles [37]. Thus, at the early stage of 250 °C–CAP, more sub-grain boundaries formed. At the same time, due to the pinning of precipitated particles, sub-grain boundaries were hindered from migrating, leading to more sub-grain structures in the 250 °C–ECAP samples.

For samples deformed at 300 °C, although dislocation slip was promoted, annihilation of dislocations was also accelerated, leading to fewer dislocation tangles and fewer nucleation sites for the secondary phase particles. Therefore, sub-grain boundaries can migrate easily due to fewer precipitated particles, resulting in larger grain sizes compared to the 250 °C–ECAP samples.

Figure 8 shows the {0001} pole figures of the 250 °C ECAP samples and 300 °C ECAP samples. In contrast with the intensive basal texture formed during extrusion, ECAP deformation generated non-basal texture components. After 1P–ECAP at 250 °C, the basal planes of most grains moved towards the shear plane (Figure 3a), which was consistent with the results of [38]. Therefore, the c-axes of these grains were approximately parallel to the normal direction of the shear plane, which caused the poles in the {0001} pole figure to rotate to a position at 45° with ND and ED, that is a 45° texture. The reason for the formation of the 45° texture may relate to slip planes of prevailing slip systems rotating towards the shear planes (SP) during ECAP [37].

Upon increasing the ECAP passes to two, the maximum intensity of the 45° texture decreased. Coarse grains with the 45° texture evolved into fine recrystallized grains with random orientations, leading to a decrease of the maximum intensity of the 45° texture. For the 250 °C–4P sample, the intensity of the 45° texture increased, mainly because basal slip was more active.

Compared with the 250°C–1P sample, the distribution of grain orientations of the 300 °C–1P sample was quite different (Figure 8d). It can be seen that the c-axes of some grains rotated towards TD, which was due to the preferred growth of grains with a TD orientation. After two passes at 300 °C, the concentration of the c-axes at ND may be related to the pyramidal slip [39]. Upon further increasing the ECAP passes to four at 300 °C, the maximum intensity was located nearly at 45° as a result of the basal slip of most grains.

### 3.2. Mechanical Properties of ECAP Samples

Figure 9 shows the tensile test results of the 1P, 2P, and 4P ECAP samples, and the corresponding values are shown in Table 2. For the 250 °C and 300 °C–ECAP samples, although the d_ave_ decreased, the corresponding yield strengths decreased. Thus, we needed to consider the change of other strengthening mechanisms that influence the yield strength, i.e., precipitation strengthening Δσ_p_, solution strengthening Δσ_p_, and dislocation strengthening Δσ_p_.

Precipitation strengthening (Δσ_p_) can be calculated through the Orowan equation [40]:(1)$$\Delta \sigma_{p}=M\frac{\mathrm{Gb}}{2\pi \sqrt{1-v}\left(\frac{0.953}{\sqrt{f}}-1\right)d_{p}}\ln\frac{d_{p}}{b}$$
where M is the Taylor factor (2.5 [41] ), G is the shear modulus (1.66 × 10^4^ MPa for Mg [41]), b is the Burgers vector (3.2 × 10^−10^ m for Mg [41]), v is Poisson’s ratio (0.35 for Mg [41]), and d_p_ and f_p_ are the average phase particles’ size and the volume fraction of particles, respectively.

Solution strengthening for the ATZ611 alloy was mainly caused by Al, Zn, and Sn solute atoms. Solution strengthening (Δσ_s_) was calculated by Equation (2) according to the Deruyttere and Gypen theoretical model [42]:(2)Δσs=kAl1nCAl+kZn1nCZn+kSn1nCSnn
where k_Al_ is 196 MPa (at.)^−2/3^ [43], k_Zn_ is 905 MPa (at.)^−2/3^ [43], k_Sn_ is 52 MPa (at.)^−2/3^ [42], and n is constant as 2/3 [44]. In this work, we assumed that ~6 wt.% Al, ~1 wt.% Zn, and ~0.9 wt.% Sn atoms were in a solid solution for the as-solutionized samples. Because of the precipitation of the secondary phase during ECAP deformation at 250 °C and 300 °C, approximately 3 wt.% Al and ~1 wt.% Zn atoms were assumed to be in the solid solution for the 250 °C sample, while ~6 wt.% Al, ~1 wt.% Zn, and ~0.9 wt.% Sn were assumed to be in the solid solution for the 300 °C sample. The values of solution strengthening were estimated to be ~31 MPa for 250 °C samples and ~40 MPa for 300 °C samples

The dislocation strengthening was calculated according to the Taylor formula:(3)Δσd=αMTGbρ12
where ρ is the dislocation density. The parameter α was 0.6 for basal–basal dislocation interaction. The dislocation density ρ of each sample was the geometric necessary dislocation density, which was obtained from the EBSD results, as shown in Table 1.

The strength increase caused by secondary-phase particles, dislocations, and solution atoms were subtracted from the measured yield strength (Table 3), and σ_0.2_ − (Δσ_p_ + Δσ_s_ + Δσ_d_) should be mainly attributed to grain boundary strengthening. From 3.1, it was found that d_ave_ decreased with the increase of ECAP passes, which should lead to an increase of grain boundary strengthening, i.e., σ_0.2_ − (Δσ_p_ + Δσ_s_ + Δσ_d_). However, as shown in Table 3, the calculated σ_0.2_ − (Δσ_p_ + Δσ_s_ + Δσ_d_) values showed a different trend, which could be related to the change of the H-P coefficient k caused by textures [45]. Texture affects k mainly by changing the proportion of different slip modes [45]. If the basal planes of most grains were parallel to the theoretical shear plane and the tensile force was along ED at the same time, it would be easier to activate basal slip, which would result in a smaller k value. Thus, the decrease in σ_0.2_ − (Δσ_p_ + Δσ_s_ + Δσ_d_) was a result of the reduction of k. Moreover, for the 300 °C-1P sample, in addition to the 45° texture, concentrated poles around TD could be observed. Basal slips were less easy for these “TD”-orientation grains during tension along the ED, resulting in a higher k value and thus the larger value of σ_0.2_ − (Δσ_p_ + Δσ_s_ + Δσ_d_).

## 4. Conclusions

The influence of ECAP deformation temperatures (250 °C and 300 °C) on the microstructure, texture, and mechanical properties of a Mg-6Al-1Zn-0.9Sn alloy was studied. The main conclusions are as follows:

1. The smallest d_ave_ of ~1.9 μm was obtained after 4P ECAP at 250 °C. Different rates of grain refinement at 250 °C and 300 °C were caused by the different amounts of precipitated secondary-phase particles.

2. The strong basal texture in as-solutionized alloy transformed to a non-basal texture during ECAP, with {0001} planes inclined ~45° to ED. Compared to the 250 °C sample, c-axes of grains in the 300 °C sample rotated to 45° at a slower rate.

3. The yield strength of the as-ECAPed alloys was significantly lower than that of the as-solutionized alloy. The formation of 45° textures facilitated the activation of basal slips, which resulted in a low value of H-P coefficient k. The yield strength of the as-ECAPed alloys decreased greatly as the K value became smaller.

## Figures and Tables

**Figure 1 materials-14-02664-f001:**
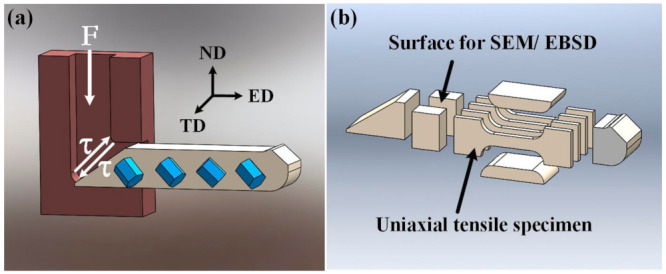
(**a**) Schematic of the ECAP process. (**b**) Samples for microstructure characterization and tensile tests.

**Figure 2 materials-14-02664-f002:**
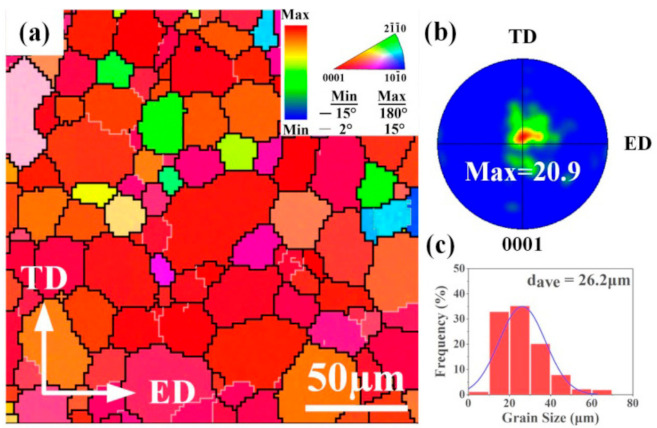
(**a**) IPF map, (**b**) {0001} pole figure, and (**c**) grain size distribution of the as-solutionized sample.

**Figure 3 materials-14-02664-f003:**
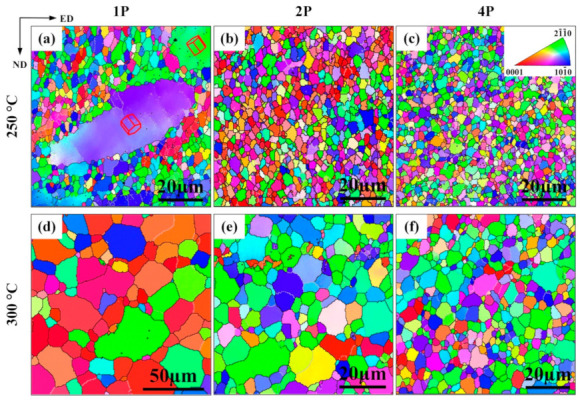
IPF maps of (**a**,**b**,**c**) 250 °C-ECAP samples and (**d**,**e**,**f**) 300 °C–ECAP samples. (**a**,**d**) 1 pass, (**b**,**e**) 2 passes, and (**c**,**f**) 4 passes.

**Figure 4 materials-14-02664-f004:**
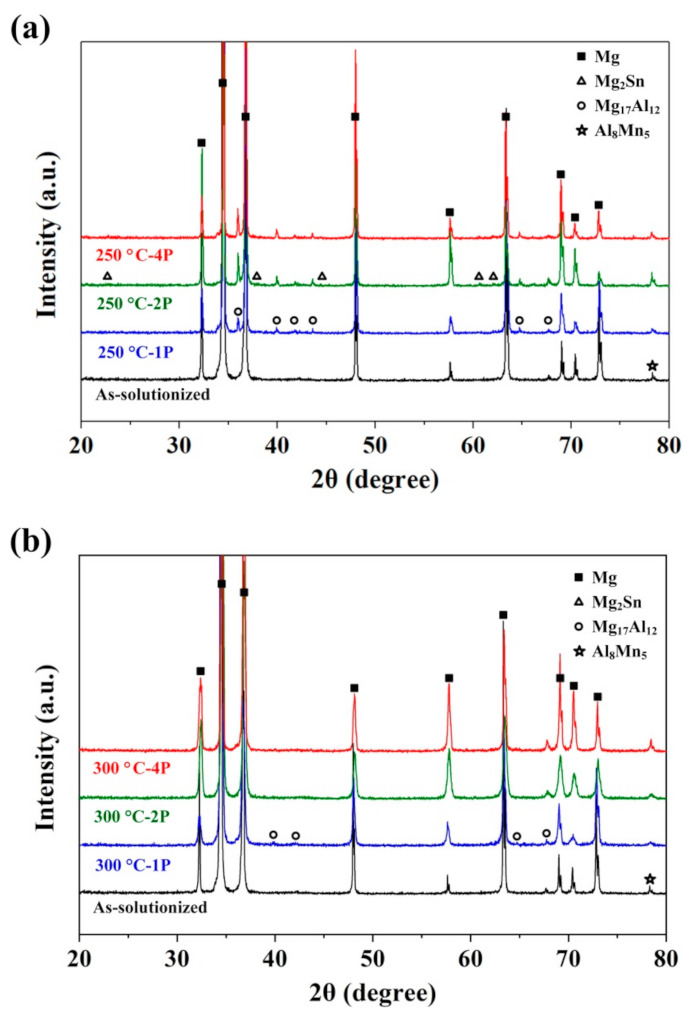
XRD results of the as-solutionized and the 250 °C and 300 °C ECAP samples. (**a**) 1 pass, 2 passes, 4 passes at 250 °C, (**b**) 1 pass, 2 passes, 4 passes at 300 °C.

**Figure 5 materials-14-02664-f005:**
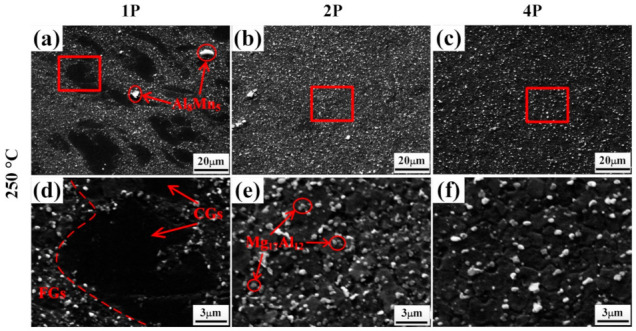
SEM images of ECAP samples deformed at 250 °C: (**a**,**d**) 1 pass, (**b**,**e**) 2 passes, and (**c**,**f**) 4 passes. (**d**–**f**) are the high-magnification images of (**a**–**c**), respectively.

**Figure 6 materials-14-02664-f006:**
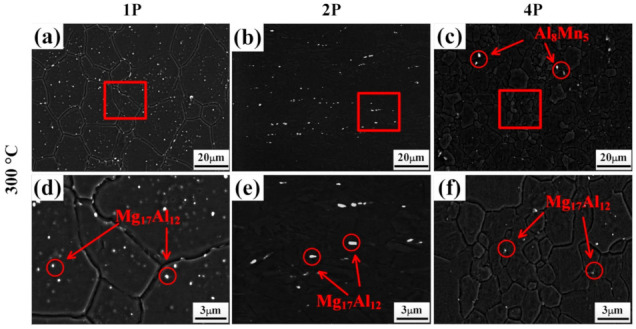
SEM images of ECAP samples at 300 °C: (**a**,**d**) 1 pass, (**b**,**e**) 2 passes, and (**c**,**f**) 4 passes. (**d**–**f**) are the high-magnification images of (**a**–**c**), respectively.

**Figure 7 materials-14-02664-f007:**
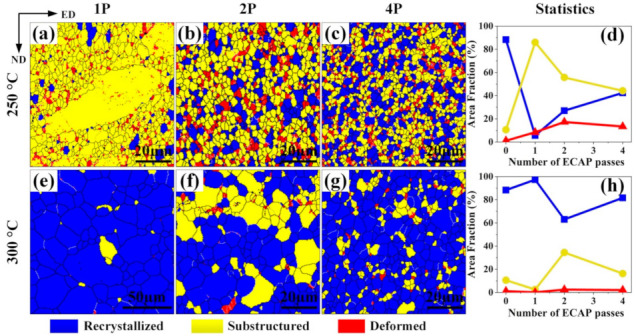
Recrystallization maps of 250 °C ECAP samples (**a**–**d**) and 300 °C ECAP samples (**e**–**h**). (**a**,**e**) One pass, (**b**,**f**) two passes, (**c**,**g**) four passes, and (**d**,**h**) the flow curves of recrystallization tendency.

**Figure 8 materials-14-02664-f008:**
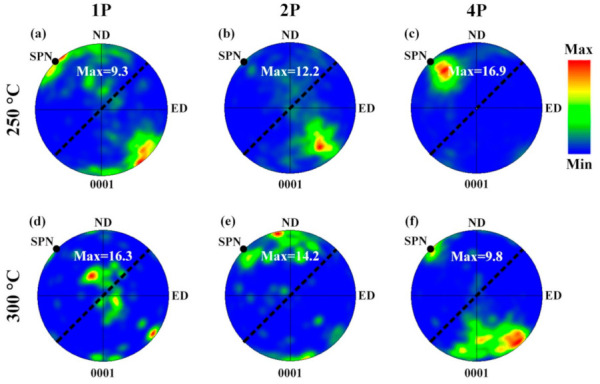
{0001} pole figure of the 250 °C–ECAP samples and (**d**–**f**) 300 °C–ECAP samples. (**a**,**d**) One pass, (**b**,**e**) two passes, and (**c**,**f**) four passes. SP, SPN are abbreviations for shear plane and shear plane normal, respectively.

**Figure 9 materials-14-02664-f009:**
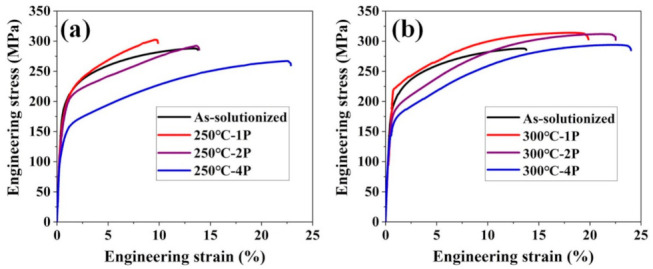
Tensile engineering stress–strain curves of as-solutionized samples and ECAPed samples. (**a**) 1 pass, 2 passes, 4 passes at 250 °C, (**b**) 1 pass, 2 passes, 4 passes at 300 °C.

**Table 1 materials-14-02664-t001:** Microstructure characteristics of the ECAP samples.

Sample	d_ave_ (μm)	d_p_ (nm)	f_p_ (%)	ρ (10^12^m^−2^)
As-solutionized	26.2	-	-	-
250 °C–1P	6.2	400	17	6.05
250 °C–2P	3.6	500	20	2.96
250 °C–4P	1.9	580	12	6.13
300 °C–1P	16.5	430	5	1.14
300 °C–2P	8.6	430	3	2.86
300 °C–4P	5.5	350	2	2.23

**Table 2 materials-14-02664-t002:** Room temperature tensile properties of samples deformed at 250 °C and 300 °C.

Sample	σ_0.2_ (MPa)	σ_b_ (MPa)	δ_p_ (%)
As-solutionized	185	288	13.4
250 °C-1P	165	302	9.5
250 °C-2P	160	293	13.4
250 °C-4P	124	267	22.6
300 °C-1P	211	314	19.4
300 °C-2P	166	312	22.1
300 °C-4P	142	294	23.6

**Table 3 materials-14-02664-t003:** Contribution of different strengthening mechanisms to the yield strength of samples deformed at 250 °C and 300 °C.

Sample	Δσ_p_ (MPa)	Δσ_s_ (MPa)	Δσ_d_ (MPa)	(Δσ_p_ + Δσ_s_ + Δσ_d_) (MPa)	σ_0.2_ − (Δσ_p_ + Δσ_s_ + Δσ_d_) (MPa)
As-solutionized	-	40	-	40	145
250 °C-1P	36.5	31	20	87.5	77.5
250 °C-2P	34.9	31	14	79.9	80.1
250 °C-4P	19.8	31	13.5	65.3	60.7
300 °C-1P	14.1	40	7.8	61.9	149.1
300 °C-2P	9.9	40	12.4	62.3	103.7
300 °C-4P	9.3	40	10	58.3	82.7

## Data Availability

Data sharing not applicable. No new data were created or analyzed in this study. Data sharing is not applicable to this article.

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
