# Peer review of "Influence of Texture on the Mechanical Properties of a Mg-6Al-1Zn-0.9Sn Alloy Processed by ECAP"

_materials, 2021, doi:10.3390/ma14102664_

Round 1

Reviewer 1 Report

The Paper is well done overall, nevertheless some revisions must be done:

  1. Spelling: there are some typing mistakes which can be easily corrected, e.g. lines: 14, 18, 59, 238, 255
  2. In the following missing or hard to understand information is listed. Please improve/change these passages accordingly:

Line 63: Describe the used Route ‘Bc’, additionally it would be beneficial to add a picture of the route i.e. Fig. 1c

Figure 3: Add the number of passes in the picture (just like the temperature) for easier understanding. à This should also be done for Figures: 5, 6, 7, 8.

Table 1: Improve the description of the table, i.e. used abbreviations. The used unit system is misleading, use [] for easier understanding. As the values from this table are later used in calculations, they should already have the appropriate unit. This is also the case for Tables 2 and 3 where units are partly missing as well.

Figure 7: Describe and define your definition of recrystallized, substructured and deformed grains in detail.

Line 160 onward: If possible, add information on the recrystallization tendency of the alloy, i.e. flow curves at the respective temperatures.

Line 187: Describe the ‘45° texture’ in more detail.

Line 206, 230: Variables are defined twice.

  1. Scientific revisions & adaptations:

For a better overall understanding of your processing scheme and Fig. 4 add a CALPHAD calculation of the used alloy, then the dependence of the existing phases on the processing temperatures can be easier discussed.

Calculations for the Hall Patch behavior of the alloy is missing. If the strengthening mechanisms for YS are discussed this should also be included. It easily manageable with the already existing data.

The as-extruded plate is mentioned multiple times – but there are no values of the as-extruded stock material in this work. Describe why it was necessary to solutionize the as-extruded plate.

Highlight how the results of your work correlates to other scientific publications on this field, i.e. texture, mechanical properties, grain size.

Add a list of abbrevations.

Reviewer 2 Report

The submitted manuscript entitled “Influence of texture on mechanical properties of a Mg-6Al-1Zn-0.9Sn alloy processed by ECAP” has some interesting results. In the work, the Authors investigated the effect of ECAP deformation temperatures at 250 °C and 300 °C on the microstructure, texture and mechanical properties of the Mg-6Al-1Zn-0.9Sn alloy. They used various research techniques to explain the grain boundary strengthening mechanism. The Authors' approach to research is a valuable asset of this work. I would like to acknowledge authors for this manuscript.

I have two questions for the Authors of the Manuscript.

  1. My question concerns the presence of zinc, can the Authors answer to which phase it segregates?
  2. Mg-Al alloys already above 2% Al show a tendency to form non-equilibrium eutectic in the structure, in this case it was not observed. How does the presence and share of Mg17Al12 phase affect the plasticity of this alloy?

Reviewer 3 Report

Dear Authors,

The topic you present is very interesting. I have a general question. 

Are photos (a - c) in Figures 5 and 6 necessary?

Kind regards

Reviewer 4 Report

This work studies the influence of deformation temperatures on grain refinement, precipitation, texture evolution, and mechanical properties of an ECAP processed Mg-6Al-1Zn-0.9Sn alloy. The deformation temperatures are 250 and 300°C. The present paper is interesting, however, to be accepted for publication the following comments need to be addressed.

  • The proficiency of the language needs a more improvement in the manuscript
  • The introduction section needs to be improved by cite new and related articles.

Otherwise, the methods and results are quite presented. The manuscript can be accepted after minor revision

Reviewer 5 Report

A large number of articles have been written on this topic. However, the research results presented in this article may be useful to some scientists. The article is well structured and does not contain unnecessary information. At the same time that I was reviewing the article, I posed several questions for the authors of this article:

1) Why were the temperatures of 250 and 300 °C chosen for the research? The article does not say about this. I think that this point needs to be justified.

2) What was heated to a temperature of 250 and 300 °C? Die? And what was the temperature of the alloy under study?

3) What devices and how did you control the temperature of the extruded alloy?

4) What was the degree of deformation during ECAP?

I believe that the answers to my questions reflected in the article allow us to improve it.

Round 2

Reviewer 1 Report

Well done on the improvements. Even if CALPHAD calculations are not neccessary - it would improve the overall structure of the work, as the discussion of microstructural behavior is easier done. Consider it for your next paper.

Reviewer 5 Report

A large number of articles have been written on this topic. However, the research results presented in this article may be useful to some scientists. The article is well structured and does not contain unnecessary information. The authors of the article gave answers to my questions. I recommend that you accept the article for publication in this version.